# A dual endosymbiosis supports nutritional adaptation to hematophagy in the invasive tick *Hyalomma marginatum*

Marie Buysse[1,2]*[†], Anna Maria Floriano[3,4][†], Yuval Gottlieb[5], Tiago Nardi[3], Francesco Comandatore[6], Emanuela Olivieri[3], Alessia Giannetto[7], Ana M Palomar[8], Benjamin L Makepeace[9], Chiara Bazzocchi[10], Alessandra Cafiso[10], Davide Sassera[3][‡], Olivier Duron[1,2][‡]

[1]MIVEGEC (Maladies Infectieuses et Vecteurs : Ecologie, Génétique, Evolution et Contrôle), Univ. Montpellier (UM) - Centre National de la Recherche Scientifique (CNRS) - Institut pour la Recherche et le Développement (IRD), Montpellier, France; [2]Centre of Research in Ecology and Evolution of Diseases (CREES), Montpellier, France, Montpellier, France; [3]Department of Biology and Biotechnology "L. Spallanzani", University of Pavia, Pavia, Italy; [4]Faculty of Science, University of South Bohemia, České Budějovice, Czech Republic; [5]Koret School of Veterinary Medicine, The Robert H. Smith Faculty of Agriculture, Food and Environment, The Hebrew University of Jerusalem, Rehovot, Israel; [6]Department of Biomedical and Clinical Sciences L. Sacco and Pediatric Clinical Research Center, University of Milan, Milan, Italy; [7]Department of Chemical, Biological, Pharmaceutical and Environmental Sciences, University of Messina, Messina, Italy; [8]Center of Rickettsiosis and Arthropod-Borne Diseases (CRETAV), San Pedro University Hospital- Center of Biomedical Research from La Rioja (CIBIR), Logroño, Spain; [9]Institute of Infection, Veterinary & Ecological Sciences, University of Liverpool, Liverpool, United Kingdom; [10]Department of Veterinary Medicine, University of Milan, Lodi, Italy

*For correspondence:
marie.buysse@ird.fr

[†]These authors contributed equally to this work
[‡]These authors also contributed equally to this work

Competing interest: The authors declare that no competing interests exist.

**Abstract** Many animals are dependent on microbial partners that provide essential nutrients lacking from their diet. Ticks, whose diet consists exclusively on vertebrate blood, rely on maternally inherited bacterial symbionts to supply B vitamins. While previously studied tick species consistently harbor a single lineage of those nutritional symbionts, we evidence here that the invasive tick *Hyalomma marginatum* harbors a unique dual-partner nutritional system between an ancestral symbiont, *Francisella*, and a more recently acquired symbiont, *Midichloria*. Using metagenomics, we show that *Francisella* exhibits extensive genome erosion that endangers the nutritional symbiotic interactions. Its genome includes folate and riboflavin biosynthesis pathways but deprived functional biotin biosynthesis on account of massive pseudogenization. Co-symbiosis compensates this deficiency since the *Midichloria* genome encompasses an intact biotin operon, which was primarily acquired via lateral gene transfer from unrelated intracellular bacteria commonly infecting arthropods. Thus, in *H. marginatum*, a mosaic of co-evolved symbionts incorporating gene combinations of distant phylogenetic origins emerged to prevent the collapse of an ancestral nutritional symbiosis. Such dual endosymbiosis was never reported in other blood feeders but was recently documented in agricultural pests feeding on plant sap, suggesting that it may be a key mechanism for advanced adaptation of arthropods to specialized diets.

## Editor's evaluation

This manuscript analyses endosymbionts harbored by field collected ticks in Spain, Italy and Israel. Using metagenomics, the authors infer the evolutionary history of a newly discovered tripartite interaction between *Hyalomma marginatum* (tick), *Franscisella*-like endosymbiont (FLE) (ancestral co-obligate endosymbiont) and a *Midichloria* symbiont (more recent co-obligate endosymbiont). This is the first description of such co-obligate symbiotic association in ticks, which usually harbor only one obligate endosymbiont.

## Introduction

Ticks evolved to hematophagy through the acquisition of key genomic adaptations (*Gulia-Nuss, 2016*; *Jia et al., 2020*) and the evolution of mutualistic interactions with microbial symbionts (*Buysse and Duron, 2021*; *Duron and Gottlieb, 2020*). Ticks rely on vertebrate blood as their sole food source throughout their development, although it is nutritionally unbalanced. Blood contains high levels of proteins, iron, and salts, but low levels of carbohydrates, lipids, and vitamins. To overcome these dietary challenges, tick genomes present large repertoires of genes related to vitamin and lipid shortages, iron management, and osmotic homeostasis (*Gulia-Nuss, 2016*; *Jia et al., 2020*). Yet, tick genomes do not contain genes for synthesis of some essential vitamins, thus, vitamin-provisioning pathways have indirectly arisen from symbiotic associations with transovarially transmitted intracellular bacteria (*Ben-Yosef et al., 2020*; *Duron et al., 2018*; *Duron and Gottlieb, 2020*; *Gerhart et al., 2016*; *Gottlieb et al., 2015*; *Guizzo et al., 2017*; *Smith et al., 2015*). This partnership allowed the long-term acquisition of heritable biochemical functions by ticks and favored their radiation into the nearly 900 tick species that currently exist (*Duron and Gottlieb, 2020*). This co-evolutionary process is reflected by microbial communities dominated by tick nutritional symbionts (*Bonnet and Pollet, 2021*; *Narasimhan and Fikrig, 2015*).

Most tick species typically harbor a single lineage of nutritional symbiont, but multiple symbiont lineages exist in different tick species (*Duron and Gottlieb, 2020*). The most common lineages belong to the *Coxiella*-like endosymbiont group and to the *Francisella*-like endosymbiont group (CLE and FLE, hereafter) (*Azagi et al., 2017*; *Binetruy et al., 2020*; *Duron et al., 2017*; *Machado-Ferreira et al., 2016*), but a few tick species may rely on other symbionts (*Duron et al., 2017*), such as the *Rickettsia*-like endosymbionts (*Narasimhan et al., 2021*) and the intramitochondrial bacterium *Midichloria* (*Sassera et al., 2006*). Once deprived of their nutritional symbionts, ticks show a complete stop of growth and moulting, as well as lethal physical abnormalities (*Ben-Yosef et al., 2020*; *Duron et al., 2018*; *Guizzo et al., 2017*; *Zhong et al., 2007*), which can be fully restored with an artificial supplement of B vitamins (*Duron et al., 2018*). Nutritional symbionts of ticks exhibit a high degree of lifestyle specialization toward mutualism. They have been found primarily in two tick organs, the ovaries and Malpighian tubules, while other organs have only been found sporadically infected (*Buysse et al., 2019*; *Duron and Gottlieb, 2020*). Ovary infection is consistent with vertical transmission of nutritional symbionts in developing oocytes. A high density of nutritional symbionts in Malpighian tubules supports a nutritional role for these symbionts, as Malpighian tubules are involved in excretion and osmoregulation. A likely hypothesis is that nutritional symbionts use tick hemolymph compounds to synthesize de novo B vitamins (*Duron and Gottlieb, 2020*).

Genomes of CLE and FLE consistently contain complete biosynthesis pathways of the same three B vitamins: biotin (vitamin $B_7$), riboflavin ($B_2$), and folate ($B_9$) (*Duron et al., 2018*; *Gerhart et al., 2018*; *Gerhart et al., 2016*; *Gottlieb et al., 2015*; *Guizzo et al., 2017*; *Nardi et al., 2021*; *Smith et al., 2015*). These three pathways form a set of core symbiotic genes essential for ticks (*Duron and Gottlieb, 2020*) while other functions are disposable. As such, the genomes of CLE, FLE, and *Midichloria* underwent massive reduction as observed in other nutritional endosymbionts of arthropods with a specialized diet (*Duron et al., 2018*; *Gerhart et al., 2018*; *Gerhart et al., 2016*; *Gottlieb et al., 2015*; *Nardi et al., 2021*; *Sassera et al., 2011*; *Smith et al., 2015*). This reduction process is primarily initiated as a consequence of relaxed selection on multiple genes which are not essential within the host stable and nutrient-rich cellular environment (*Bennett and Moran, 2015*; *McCutcheon et al., 2019*). Over time, genes encoding for functions such as motility, regulation, secondary-metabolite biosynthesis, and defense are pseudogenized and then definitively jettisoned (*Bennett and Moran, 2015*; *McCutcheon et al., 2019*).

Coevolution of hosts and their nutritional symbionts over millions of years is expected to result in stable co-cladogenesis (*Bennett and Moran, 2015*). However, in sap-feeding insects where symbionts compensate for nutritional deficiencies of the sap diet, cases of unstable interactions have been observed (*Bennett and Moran, 2015*; *McCutcheon et al., 2019*). Indeed, as the symbiont genome decays, repair mechanisms can be lost, leading to faster pseudogenization, which can affect genes in all functional categories including those involved in nutritional supplementation (*Bennett and Moran, 2015*; *McCutcheon et al., 2019*). This may ultimately lead to overly reduced and maladapted genomes, limiting beneficial contributions to hosts and opening the road to wholesale replacement by new symbiont(s) (*Bennett and Moran, 2015*; *McCutcheon et al., 2019*; *Sudakaran et al., 2017*). This degeneration–replacement model has been proposed in sap-feeding insects (*Campbell et al., 2015*; *Lukasik et al., 2018*; *Manzano-Marın et al., 2020*; *Matsuura et al., 2018*), but is difficult to document as such replacements are expected to be transient (*McCutcheon et al., 2019*). In parallel, other forms of multi-partite interactions have been reported, where a novel bacterium with similar function to a pre-existing symbiont enters a bipartite system (*Husnik and McCutcheon, 2016*; *Santos-Garcia et al., 2018*; *Takeshita et al., 2019*). This can lead to relaxation of conservative selection pressure on the original symbionts, which can lose genes that were previously fundamental, ultimately leading to a tripartite stable symbiosis (*Husnik and McCutcheon, 2016*; *Takeshita et al., 2019*).

In this study, we document a complex nutritional endosymbiosis in the invasive bont-legged tick *Hyalomma marginatum* characterized by a maladapted ancestral symbiont genome compensated by gene functions of a more recently acquired symbiont. This large tick species usually feeds on humans and a diversity of mammals and birds in the Mediterranean climatic zone but has recently expanded its range to the north, invading Western Europe (*Vial et al., 2016*). *Hyalomma marginatum* can transmit Crimean-Congo hemorrhagic fever virus, an emerging disease affecting humans in the northern Mediterranean Basin, as well as protozoan parasites and intracellular bacterial pathogens such as *Rickettsia aeschlimannii* (*Vial et al., 2016*). Previous studies detected the presence of diverse highly prevalent bacterial symbionts across *H. marginatum* populations (*Di Lecce et al., 2018*), but their contribution to tick nutrition was not studied. A previous study showed that high densities of FLE can be observed in Malpighian tubules and ovaries of *H. marginatum*, while infections in other organs are scattered (*Azagi et al., 2017*). Here, we sequenced three microbial metagenomes of *H. marginatum* in three distinct locations, and determined the nutritional interactions in this unique symbiosis system.

## Results

### Microbial metagenomes

We sequenced three microbial metagenomes from *H. marginatum* specimens collected in Spain (i.e., the Hmar-ES metagenome), Italy (Hmar-IT), and Israel (Hmar-IL). We obtained 57,592,510 paired-end reads from the Hmar-ES metagenome, 68,515,944 paired-end reads from Hmar-IT, and 10,807,740 paired-end reads from Hmar-IL. We assembled one FLE genome with >99% estimated completeness from each metagenome (FLE-Hmar-ES, FLE-Hmar-IT, and FLE-Hmar-IL; also designated as FLE-Hmar genomes; *Supplementary file 1a*). In parallel, we assembled one genome with >95% estimated completeness from each metagenome for the bacterium *Midichloria*, and thus obtained three additional complete genomes (Midi-Hmar-ES, Midi-Hmar-IT, and Midi-Hmar-IL; also designated as Midi-Hmar genomes; *Supplementary file 1a*). In addition, we detected a third bacterium, the tick-borne pathogen *R. aeschlimannii*, in the Hmar-IT and Hmar-IL, but not Hmar-ES, metagenomes, and further assembled one genome with 94% estimated completeness (RAES-Hmar-IT) and one partial (RAES-Hmar-IL) genome (*Supplementary file 1a*; *Figure 1*).

The assemblies of the three FLE draft genomes (13 contigs for FLE Hmar-IT, 17 for FLE Hmar-ES, and 25 FLE Hmar-IL) were very similar in size (1.51–1.536 Mb), average G+C content (31.11–31.23%), number of predicted protein-coding genes (937–961), and protein-coding sequences with inactivating mutations (pseudogenes, 789–798), no prophages, and two identical insertion sequences (*Supplementary file 1a*). Phylogenomic analysis based on 436 single-copy orthologs (SCOs) *Francisella* genes showed that FLE-Hmar-ES, FLE-Hmar-IT, and FLE-Hmar-IL cluster together in a robust clade within other FLE of ticks (*Figure 2A*). The closest relative of this clade is the FLE of another *Hyalomma* species, *Hyalomma asiaticum* from China, while FLEs of other tick genera (*Ornithodoros*, *Argas*, and *Amblyomma*) are more distantly related. The FLE clade is embedded

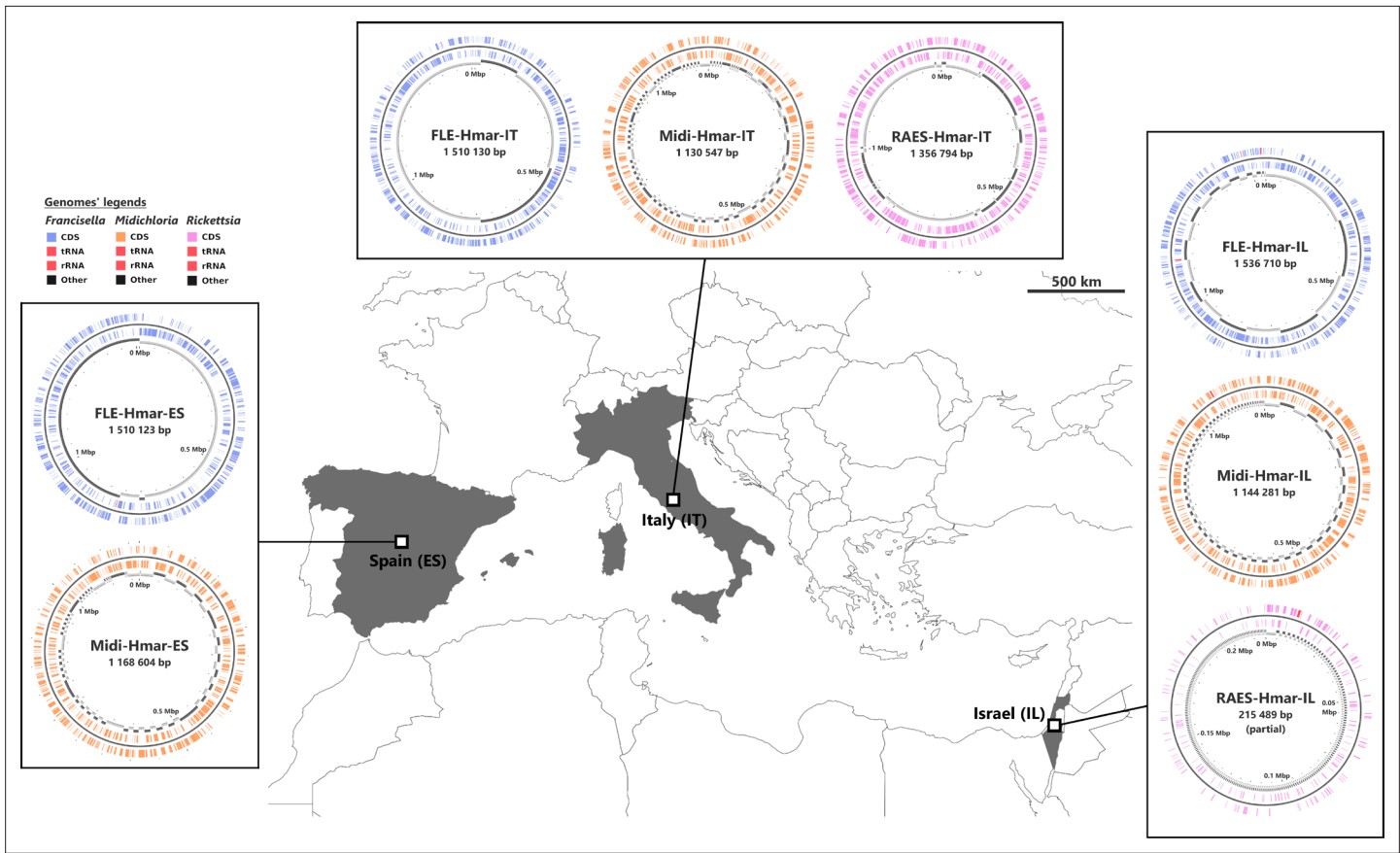

**Figure 1.** Sampled locations for *Hyalomma marginatum* ticks. Sampling countries are colored in gray. The symbiont genomes retrieved from each site are graphically represented in blue, orange, and pink, corresponding to *Francisella*-LE (FLE), *Midichloria* (Midi), and *R. aeschlimannii* (RAES), respectively. Circles on genome maps correspond to the following: (1) forward strand genes; (2) reverse strand genes; (3) in gray and black, contigs. Circular maps of the genomes were produced using CGView.

The online version of this article includes the following figure supplement(s) for figure 1:

**Figure supplement 1.** *Rickettsia* phylogenetic tree assessing the affiliation of the *Rickettsia* detected in *Hyalomma marginatum* specimens from Italy and Israel.

**Figure supplement 2.** Phylogenetic relationship of the *Hyalomma* species used in our study.

**Figure supplement 3.** Presence of FLE and *Midichloria* within the *Hyalomma* genus.

in the *Francisella* genus and shares a common origin with pathogenic species, including the tularemia agent, *F. tularensis*, and an opportunistic human pathogen, *F. opportunistica* (*Figure 2A*). The FLE-Hmar genomes are similar in organization and content overall, apart from a 127,168 bp inversion in FLE-Hmar-ES and a 4165 bp inversion in FLE-Hmar-IL (*Figure 2B*). Compared to other FLE genomes (i.e., FLEOm and FLEAm), they are slightly reduced in size and exhibit substantial rearrangements (*Figure 2B*, *Supplementary file 1a and b*). Based on gene orthologs, the coregenome of these FLE shares 743 genes, while the core genome of the FLE of *H. marginatum* contains 796 genes because the FLE-Hmar genomes share 53 genes that are not present in other FLE genomes (*Figure 2C*). Interestingly, pseudogenes are highly abundant in the three genomes of FLE of *H. marginatum,* compared with other FLE genomes (*Figure 2D*), and most functional categories are impacted by this advanced pseudogenization process. Most of the genes associated with virulence in pathogenic *Francisella* species, including the *Francisella* pathogenicity island (FPI) and secretion system, are pseudogenized or completely missing in FLE-Hmar genomes, consistent with previous observations in other FLEs (*Figure 4—figure supplement 1*). Furthermore, FLE-Hmar genomes contain fewer genes involved in replication, recombination, and repair than any other FLEs analyzed (*Figure 2C*). The nucleotide excision repair system (*uvrABCD*) in FLE-Hmar genomes is conserved; however, the DNA mismatch repair system (*mutSL*) is fully pseudogenized.

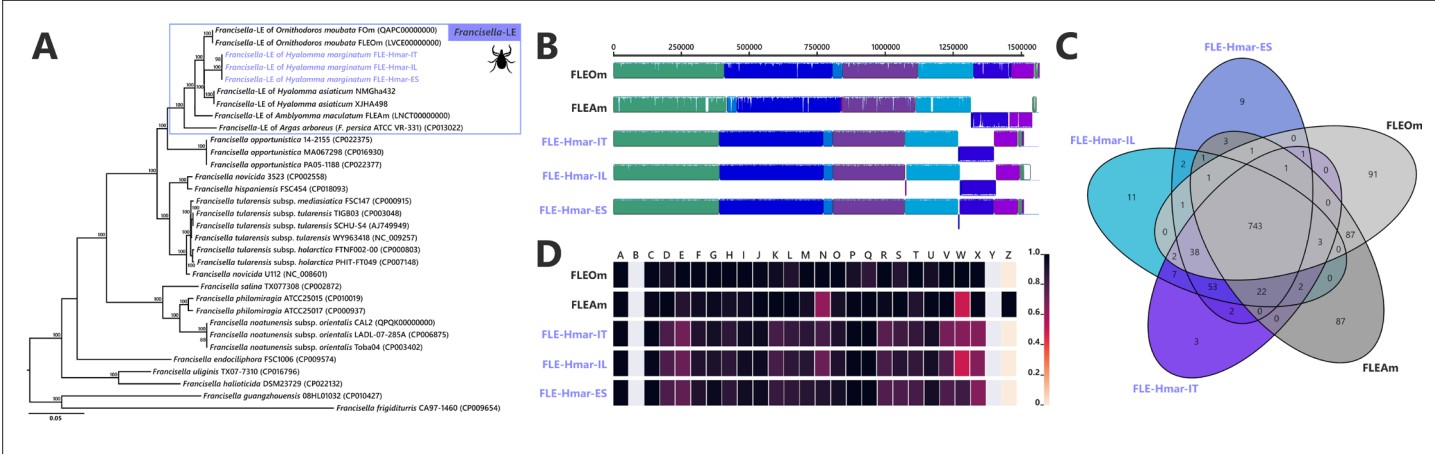

**Figure 2.** Comparative genomic features of FLE. (**A**) Whole-genome phylogenetic relationship of the three FLE genomes obtained from *Hyalomma marginatum* specimens (highlighted), FLE of other tick species, and *Francisella* pathogenic species, including *F. tularensis*. The phylogenetic tree was inferred using maximum likelihood from a concatenated alignment of 436 single-copy orthologs genes (132,718 amino acids; best-fit approximation for the evolutionary model: CPREV+G4+I). The numbers on each node represent the support of 1000 bootstrap replicates; only bootstrap values >70% are shown. The scale bar is in units of substitution/site. Sequences from newly sequenced FLE genomes are indicated by a blue font. (**B**) Whole-genome synteny of the three FLE genomes obtained from *H. marginatum* specimens (blue font) and representative FLE (black font). Each contiguously colored locally collinear block (LCB) represents a region without rearrangement of the homologous backbone sequence. LCBs below the center in *Francisella* genomes represent blocks in the reverse orientation. The height of color bars within LCBs corresponds to the average level of similarity (conservation) profile in that region of the genome sequence. Areas that are completely white were not aligned and contain sequence elements specific to a particular genome. (**C**) Venn diagram representing orthologs distribution between the three FLE genomes obtained from *H. marginatum* specimens (blue font) and representative FLE (black font). (**D**) Clusters of orthologous groups of proteins (COGs) comparisons between the three FLE genomes obtained from *H. marginatum* specimens (blue font) and representative FLE (black font). The color scale indicates the percentage of functional genes per category. Gray color means that genes are not present in a given category. The COG category names (capital letters from A to Z) correspond to the NCBI reference naming system.

In other FLEs, these systems are conserved, with the exception of the *mutSL* system in the FLE of *Ornithodoros moubata* and *H. asiaticum* genomes (**Figure 4—figure supplement 2**).

The *Midichloria* genomes, Midi-Hmar-ES, Midi-Hmar-IT, and Midi-Hmar-IL, were assembled in 179, 359, and 123 contigs, respectively, and their genomes reach 1.13–1.16 Mb, with consistent G+C content (35.03–35.23%), number of predicted protein-coding genes (1032–1071), pseudogenes (190–192), no prophages, and between three and 10 insertion sequences (**Supplementary file 1a**). Remarkably, their average coding density (76%) is higher than those of FLEs of *H. marginatum* (54%), and furthermore, those of other FLE representatives (range from 57% to 69%). Phylogenomic analysis based on 242 SCOs *Rickettsiales* genes shows that Midi-Hmar-ES, Midi-Hmar-IT, and Midi-Hmar-IL cluster in a highly supported monophyletic sister group of the single other sequenced *Midichloria* genome, *Candidatus* M. mitochondrii strain IricVA (*M. mitochondrii* hereafter), the intra-mitochondrial endosymbiont of the European sheep tick, *Ixodes ricinus*. Altogether, they delineate a monophyletic clade within the *Midichloriaceae* family (**Figure 3A**). The high number of Midi-Hmar contigs precluded comparison of their genome structures. However, gene contents of the Midi-Hmar genomes are highly similar, differing substantially from *M. mitochondrii* (**Supplementary file 1a and b**). Based on gene orthologs, the pan-genome of *Midichloria* shared 745 genes, and Midi-Hmar genomes shared 181 genes that are not present in *M. mitochondrii* (**Figure 3B**). Pseudogenization only impacted a few functional categories (e.g., genes of chromatin structure and dynamics [category B] and genes of the mobilome [prophages and transposons, category X]), across Midi-Hmar genomes (**Figure 3C**). By contrast to FLE genomes, the DNA mismatch repair system (*mutSL*) and the nucleotide excision repair (*uvrABCD*) are not pseudogenized in Midi-Hmar genomes and are thus potentially functional (**Figure 4—figure supplement 2**).

In two metagenomes, we also detected contigs belonging to *R. aeschlimannii*. We recovered one possibly complete chromosome (1.35 Mb for 165 contigs, with 1122 genes, 475 pseudogenes, and 32.32% G+C content) for RAES-Hmar-IT (**Supplementary file 1a**), whereas for RAES-Hmar-IL only a partial genome (0.215 Mb) was obtained. No *Rickettsia* reads were detected in the Hmar-ES

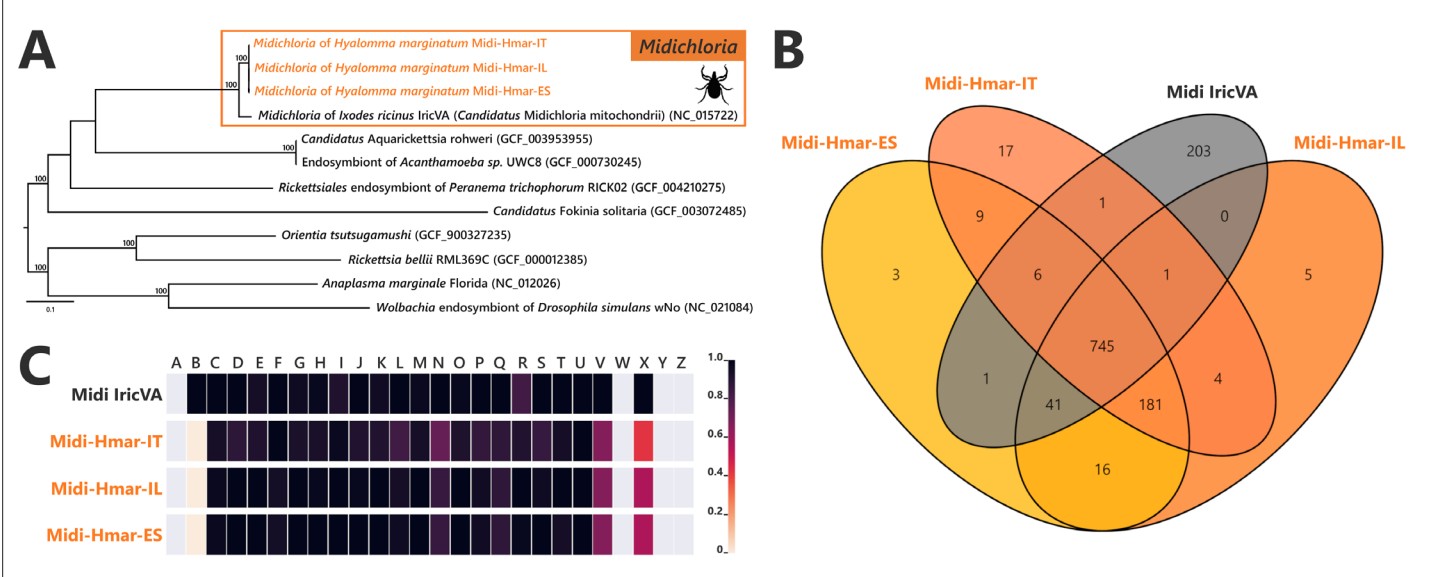

**Figure 3.** Comparative genomic features of *Midichloria*. (**A**) Whole-genome phylogenetic relationship of the three *Midichloria* genomes obtained from *Hyalomma marginatum* specimens (highlighted), '*Ca* M. mitochondrii', and other Rickettsiales species. The phylogenetic tree was inferred using maximum likelihood from a concatenated alignment of 242 single-copy orthologs genes (49,957 amino acids; best-fit approximation for the evolutionary model: LG+G4+I). The numbers on each node represent the support of 1000 bootstrap replicates; only bootstrap values >70% are shown. The scale bar is in units of substitution/site. Sequences from newly sequenced *Midichloria* genomes are indicated by an orange font. (**B**) Venn diagram representing orthologs distribution between the three *Midichloria* genomes obtained from *H. marginatum* specimens (orange font) and '*Ca*. M. mitochondrii' (black font). (**C**) Clusters of orthologous groups of proteins (COGs) comparisons between the three *Midichloria* genomes obtained from *H. marginatum* specimens (orange font) and '*Ca*. M. mitochondrii' (black font). The color scale indicates the percentage of functional genes per category. Gray color means that genes are not present in a given category. The COG category names (capital letters from A to Z) correspond to the NCBI reference naming system.

metagenome. This may be due to the absence of the bacterium in the Hmar-ES microbiome, or, possibly, to experimental artifact. This metagenome was obtained by sequencing DNA extracted from the dissected ovaries of 10 females, whereas the Hmar-IT metagenome was sequenced from DNA extracted from a whole female, suggesting that *Rickettsia* may be absent or at low density in tick ovaries. This pattern of tissue localization may be corroborated by the Hmar-IL metagenome, sequenced from a pool of dissected ovaries of 10 ticks, but for which we obtained only a few *Rickettsia* reads. This reflects either the absence of *Rickettsia* in the dissected ticks (*R. aeschlimannii* usually reaches low-to-medium prevalence in field populations of *H. marginatum*; *Azagi et al., 2017*), or a lower abundance of *Rickettsia* in ovaries. Phylogenetic analysis based on 324 SCOs genes present in *Rickettsia* genomes showed that RAES-Hmar-IT clusters with the tick-associated human pathogen *R. aeschlimannii* (including the strain MC16 previously isolated from *H. marginatum*) and altogether delineate a robust clade with the Spotted Fever *Rickettsia* group (*Figure 1—figure supplement 1*). Overall, the low abundance of *Rickettsia* in the Hmar-IL metagenome (8× average genome coverage), its absence in the Hmar-ES metagenome, as well as the genetic proximity with *R. aeschlimannii,* are all indicative of a tick-borne pathogen, rather than a nutritional endosymbiont.

## Complementation in nutritional endosymbiosis

While all previously sequenced FLE genomes contain complete pathways for biotin, riboflavin, and folate, the FLE-Hmar genomes only retain intact pathways for riboflavin and folate (*Figure 4A*). The biosynthesis pathway for biotin is non-functional, as from six genes of the complete pathway, five (*bioA*, *bioD*, *bioC*, *bioF*, and *bioB*) are pseudogenized in the FLE-Hmar genomes (only *bioH* is not pseudogenized; *Figure 4A*). Pseudogenization is due to indels or mutations that induce premature codon stops interrupting the coding sequences, indicating that FLE-Hmar cannot produce biotin (*Figure 4—figure supplement 3*). While *bioA*, *bioD*, *bioC*, and *bioF* contains multiple indels or mutations (introducing premature stop codon, or missing start codon), *bioB* shows a low level of pseudogenization with a unique indel. A stop codon comes from a reading frame shift due to the insertion

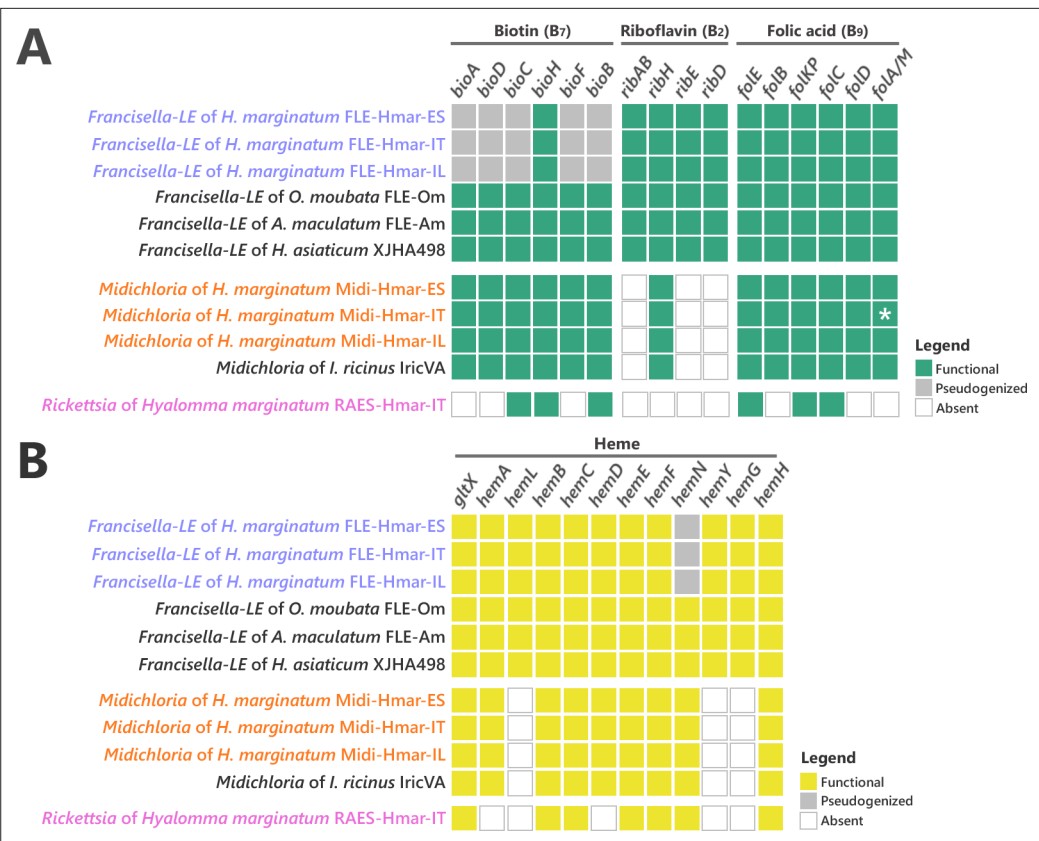

**Figure 4.** Conservation level of biosynthetic pathways involved in the symbiotic interaction between ticks and endosymbionts. (**A**) Biosynthetic pathways for the three key B vitamins (biotin, folic acid, and riboflavin) required for tick nutrition of FLE, *Midichloria*, and *Rickettsia*. Newly sequenced FLE, *Midichloria*, and *Rickettsia* genomes are indicated by a blue, an orange, and a pink font, respectively. Green squares, functional genes; gray squares, pseudogenes; white squares, missing genes; asterisk, partial gene sequence that do not contain internal codon stop but that are located at the beginning of a contig. (**B**) Heme biosynthetic pathway of FLE, *Midichloria*, and *Rickettsia*. Newly sequenced FLE, *Midichloria*, and *Rickettsia* genomes are indicated by a blue, an orange, and a pink font, respectively. Yellow squares, functional genes; gray squares, pseudogenes; white squares, missing genes.

The online version of this article includes the following figure supplement(s) for figure 4:

**Figure supplement 1.** Conservation level of the genes involved in the *Francisella* pathogenicity island (FPI) on FLE of *Hyalomma marginatum* and references genomes.

**Figure supplement 2.** Conservation level of the genes involved in the mismatch repair system (*mutSL*) and the nucleotide excision repair system (*uvrABCD*) on FLE and *Midichloria* of *Hyalomma marginatum* and references genomes.

**Figure supplement 3.** Pseudogenization of biotin synthesis genes.

**Figure supplement 4.** Conservation level of (**A**) pantothenic acid, nicotinic acid, pyroxidine, and thiamine B vitamins and (**B**) cofactors biosynthetic pathways of FLE, *Midichloria*, and *Rickettsia*.

**Figure supplement 5.** Phylogenetic trees of representative genes of riboflavin and folate biosynthesis pathways of FLE and *Midichloria*.

of one adenine base on a poly(A) zone (*Figure 4—figure supplement 3*). Since infidelity at poly(A) tracks can rescue pseudogenes in endosymbionts (*Tamas et al., 2008*), the *bioB* gene could be thus functional. Partial biosynthesis pathways of four other B vitamins (pantothenic acid: $B_5$; nicotinic acid: $B_3$; pyridoxine: $B_6$; and thiamine: $B_1$), each containing one to four pseudo/missing genes, are also present in the FLE-Hmar genomes, a pattern consistent with other FLE genomes (*Figure 4—figure supplement 4*).

The B-vitamin pathways in Midi-Hmar genomes are similar to those of *M. mitochondrii*. A complete pathway of biotin was consistently recovered in *Midichloria* genomes (*Figure 4A*). For folate, five

of the six biosynthesis genes (*folE*, *folB*, *folKP*, *folC,* and *folD*) were present but the sixth gene, *folA* (encoding for a dihydrofolate reductase), was not. However, we detected an alternative dihydrofolate reductase gene, *folM* (also known as *ydgB*), that was shown in other bacteria to provide the same metabolic function as *folA* (*Giladi et al., 2003*) suggesting that the folate pathway is functional in Midi-Hmar genomes. Only partial pathways were detected for riboflavin and pantothenic acid with three and four missing genes, respectively (*Figure 4A* and *Figure 4—figure supplement 4*).

No B vitamin pathway seems complete in the RAES-Hmar genomes. In the RAES-Hmar-IT genome, partial pathways of biotin and folate were detected with each three missing genes (*Figure 4A*). For pantothenic acid, only one of the five biosynthesis genes is present (*Figure 4—figure supplement 4*). No genes for other B vitamin pathways were found in the RAES-Hmar-IT genome. No additional B vitamin gene was observed in the other *Rickettsia* genome, RAES-Hmar-IL, which is incomplete. There is thus no obvious evidence that presume that RAES-Hmar is B-vitamin provisioning endosymbionts in *H. marginatum.*

We further searched for B vitamin genes of other origins than FLE, *Midichloria,* and *Rickettsia* genomes in tick metagenomes. In the Hmar-ES and Hmar-IL metagenomes (obtained from ovary samples), we found no significant hits, indicating that neither an additional B vitamin provisioning microorganism was present, nor did the tick genome itself encodes B vitamin pathways. In the Hmar-IT metagenome (obtained from a whole tick), we found four biotin genes (*bioA*, *bioD*, *bioF*, and *bioB*) in low abundance contigs of *Staphylococcus* bacteria. Previous studies report the presence of *Staphylococcus*, a common skin-associated bacteria, on tick exoskeletons and midguts and hypothesize this to be due to contamination, or acquisition during feeding (*Greay et al., 2018*).

Ticks differ from other eukaryotes in that they lack most of the genes encoding proteins required to produce and degrade heme (*Gulia-Nuss, 2016*; *Jia et al., 2020*). We thus searched for relevant pathways in the symbiont genomes and found that the FLE-Hmar and Midi-Hmar genomes contain genes involved in heme and iron homeostasis. The heme biosynthetic pathway was consistently

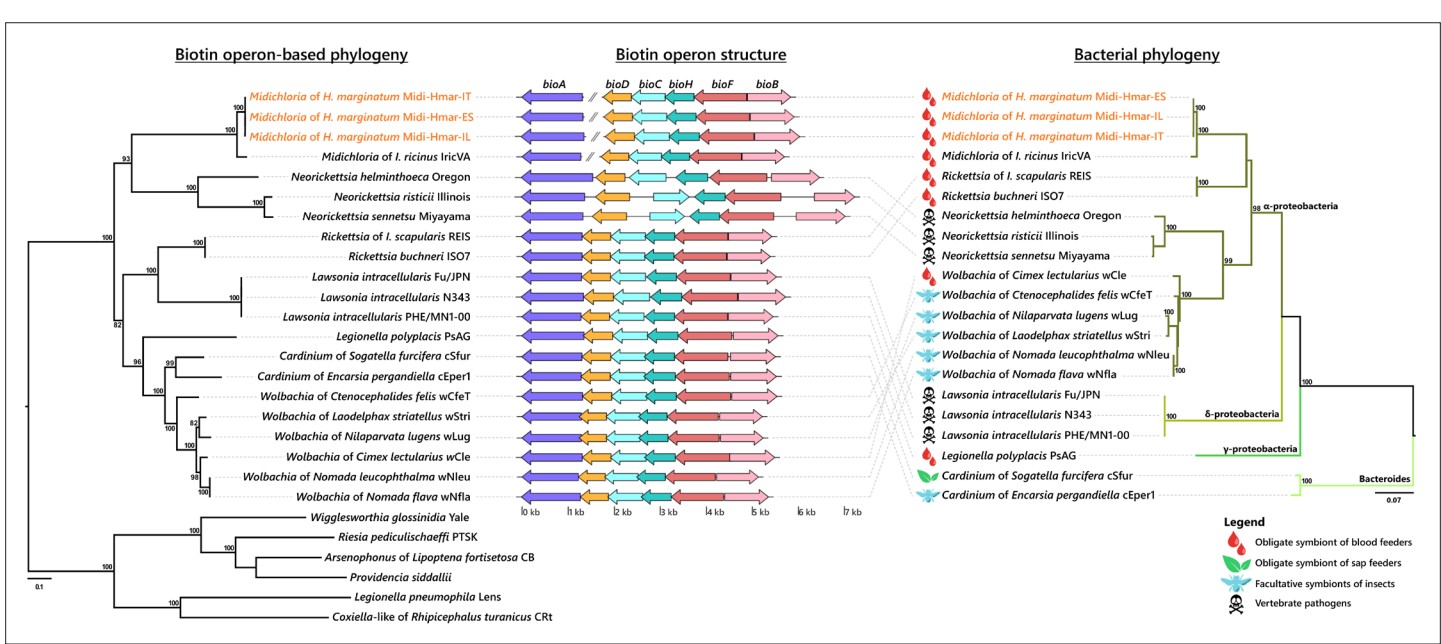

**Figure 5.** Evolutionary relationships (left) and structure (middle) of the streamlined biotin operon, confronted to the bacterial phylogeny (right). The phylogenetic tree of the streamlined biotin operon (left) was inferred using maximum likelihood (ML) from a concatenated alignment of the six genes composing the biotin biosynthetic pathway (1493 amino acids; best-fit approximation for the evolutionary model: CPREV+G4+I). The numbers on each node represent the support of 1000 bootstrap replicates; only bootstrap values >70% are shown. The scale bar is in units of substitution/site. Sequences from newly sequenced *Midichloria* genomes are indicated by an orange font. Structure of the streamlined biotin operon (middle) is represented by arrows in which each colored arrow corresponds to an intact gene and its direction. A discontinuous line corresponds to genes encoding on two separated contigs. The bacterial phylogenetic tree (right) was inferred using ML from an alignment of 16S rDNA sequences (1362 bp unambiguously aligned; best-fit approximation for the evolutionary model: GTR+G+I). The numbers on each node represent the support of 1000 bootstrap replicates; only bootstrap values >70% are shown. The scale bar is in units of substitution/site. Sequences from *Midichloria* of *Hyalomma marginatum* are indicated by an orange font.

detected, as well as a gene encoding heme oxygenase that catalyzes the degradation of heme and the subsequent release of iron (*Figure 4B*). In these genomes, we detected similar genetic patterns as those observed in the other FLE and *Midichloria* genomes. However, the heme pathway is partially degraded in FLE-Hmar genomes since their *hemN* gene is pseudogenized, a pattern not observed in other FLE genomes, suggesting an early stage of degradation for the heme pathway. This gene is present and not pseudogenized in the Midi-Hmar genomes (*Figure 4B*), suggesting these symbionts may compensate for the FLE heme pathway. In addition, three other genes, *hemL*, *hemY*, and *hemG* were not found in *Midichloria* genomes, but were present in FLE genomes (*Figure 4B*). As a result, a combination of FLE and *Midichloria* genes may be required to obtain a complete heme pathway.

## Phylogenetic origins of B vitamin biosynthetic pathways

Our analysis shows that the B vitamin biosynthesis genes in the FLE-Hmar genomes have all been vertically inherited from a *Francisella* ancestor as their B vitamin genes are always more closely related to *Francisella* orthologous genes than to genes of more distantly related bacteria (examples are shown in *Figure 4—figure supplement 5A-B*). Similarly, folate and riboflavin biosynthesis genes in the Midi-Hmar genomes are of *Midichloria* origin as these genes are more closely related to orthologous genes present in *M. mitochondrii* and other members of the *Midichloriaceae* family than to genes of other bacteria (*Figure 4—figure supplement 5A-B*).

By contrast, the six biotin genes present in the Midi-Hmar genomes did not originate from the *Midichloriaceae* family, but from an exogenous origin. The biotin synthesis genes *bioD*, *bioC*, *bioH*, *bioF*, and *bioB* formed a compact operon in the Midi-Hmar genomes while the *bioA* gene is more distantly located along the bacterial chromosome (*Figure 5*). A similar operon structure is also present on the *M. mitochondrii* genome. In contrast, an arrangement in which the *bioA* is gene physically associated with the remainder of the biosynthetic operon was identified on the genomes of multiple intracellular bacteria that are all distantly related to *Midichloria,* including the *Cardinium* endosymbionts (Bacteroidetes: Flexibacteraceae) of parasitoid wasps and leaf-hoppers; the *Wolbachia* endosymbionts (Alphaproteobacteria: Anaplasmataceae) of bed bugs, bees, planthoppers, and fleas; the *Legionella* endosymbiont (Gammaproteobacteria: Legionellaceae) of the rodent lice; *Rickettsia buchneri* (Alphaproteobacteria: Rickettsiaceae) symbiont of the blacklegged tick *Ixodes scapularis*; and diverse

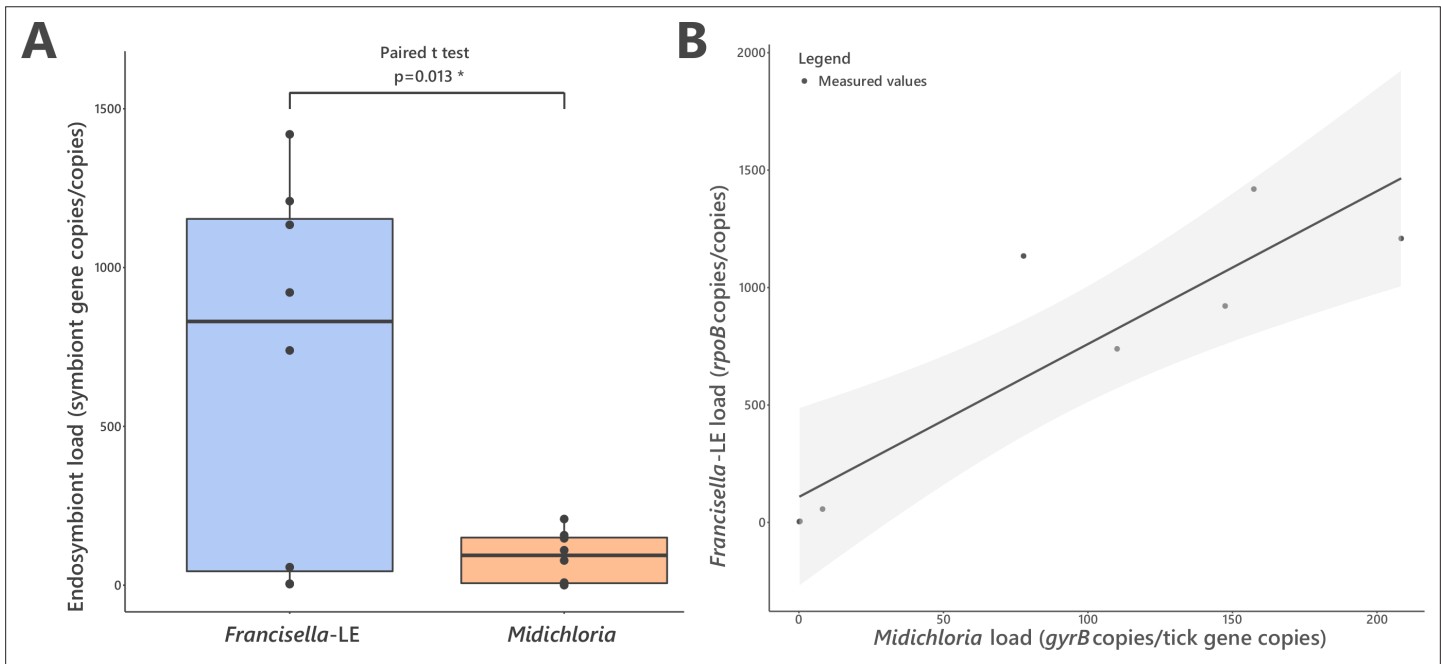

**Figure 6.** FLE and *Midichloria* loads in ovaries of *Hyalomma marginatum*. (**A**) Representation and statistical comparison of FLE and *Midichloria* densities. (**B**) FLE density as a function of *Midichloria* density in *H. marginatum* ovaries. Gray area represents 95% confidence interval.

The online version of this article includes the following source data for figure 6:

**Source data 1.** FLE and *Midichloria* densities in *H. marginatum* ovaries.

intracellular pathogens of vertebrates, such as *Lawsonia* (Deltaproteobacteria: Desulfovibrionaceae) and *Neorickettsia* (Alphaproteobacteria: Anaplasmataceae) (*Figure 5* and *Supplementary file 1c*). In most cases, the biotin genes form a compact and streamlined operon of 5–7 kb without major structural variation among bacteria. Phylogenetic analyses of the six biotin synthesis genes consistently exhibited similar evolutionary patterns (*Figure 5*). These patterns suggest that the biotin synthesis genes were acquired as a whole operon by an ancestor of *Midichloria* from an unrelated intracellular bacterium. In some cases, this streamlined biotin operon is also associated with nutritional symbionts of obligate blood feeders, such as *Wolbachia* of bedbugs, *Legionella* of rodent lice, and *R. buchneri* in the blacklegged tick, *I. scapularis* (*Figure 5*).

## Co-infections of nutritional endosymbionts in tick ovaries

We next assessed the abundance of FLE and *Midichloria* in *H. marginatum* ovaries, which are the key organ where vertical transmission of nutritional symbionts to developing oocytes occurs (*Azagi et al., 2017*). Real-time qPCR assays on 10 field *H. marginatum* adult females indicated that FLE and *Midichloria* are both abundant in tick ovaries, with FLE and *Midichloria* detected in 10 and 9 ovarian samples, respectively. High densities were always observed for FLE (686.09±206.89 FLE *rpoB* gene copies per tick *cal* gene copies) and for *Midichloria* (88.72±28.41 *Midichloria gyrB* gene copies per tick *cal* gene copies) (*Figure 6A*). FLE were consistently most abundant with densities 5.8–21.4× higher than *Midichloria* densities (paired t-test, p=0.013; one outlier was removed from analysis, n=8). The FLE and *Midichloria* densities covaried positively in individual ticks ($\chi$2=23.937, p=0.003; *Figure 6B*). FLE and *Midichloria* densities are provided in *Figure 6—source data 1*.

## Evolutionary history of endosymbiosis in the *Hyalomma* genus

We further examined the evolutionary history of *Hyalomma* species and their associations with FLE and *Midichloria* through a bibliographic survey. Out of 10 *Hyalomma* species with sufficient published data, all were found infected by FLE, and 9 by *Midichloria* (*Supplementary file 1d*). We next reconstructed *Hyalomma* phylogeny using mitochondrial DNA sequences of the cytochrome c oxidase I (*cox1*) gene available on Genbank (*Figure 1—figure supplement 2*). Mapping of FLE and *Midichloria* onto the *Hyalomma* phylogenetic tree showed that co-infection by FLE and *Midichloria* is widespread across all *Hyalomma* lineages (*Figure 1—figure supplement 3*). However, FLE and *Midichloria* co-infection was not detected in one species, *H. asiaticum*, nested within the *Hyalomma* phylogeny among co-infected species.

## Discussion

In this study, we show that the tick *H. marginatum* is a complex symbiotic system, combining genes of diverse phylogenetic origins to generate nutritional adaptations to obligate hematophagy. This tick harbors a microbiome functionally analogous to those observed in other tick species, dominated by intracellular bacterial symbionts producing the core set of B vitamins (biotin, riboflavin, and folate) essential for tick growth and survival. However, while in other tick species, as in other obligate blood feeders, production of all three core B vitamins is performed by a single nutritional symbiont (*Binetruy et al., 2020*; *Duron et al., 2017*; *Duron and Gottlieb, 2020*), *H. marginatum* harbors two bacterial partners whose core B vitamins biosynthetic capacities depend on a combination of genes, with indigenous and exogenous phylogenetic origins. This multi-partner nutritional symbiosis is stably maintained at least at the tick species level, as supported by our founding of genetically similar FLE and *Midichloria* present in *H. marginatum* populations from geographically distant regions.

Genome reconstructions confirmed the key role of symbionts in *H. marginatum* nutrition, but contrary to other tick species, no single symbiont does compensate for all nutritional needs. The FLE-Hmar genomes contain the genes to synthesize folate and riboflavin but its biotin synthesis pathway is degraded, and thus non-functional. Genome reduction is a conserved trait in all FLEs, but this process has extended in the FLE of *H. marginatum*, with an additional loss of genes in most COG functional categories. This process could have been triggered by the pseudogenization of the mismatch repair system (*mutSL*), leaving the FLE of *H. marginatum* with only one DNA repair system, nucleotide excision repair (*uvrABCD*). Absence of the *mutSL* system, coupled with strict clonality and a small population size during transovarial transmission, have probably led the FLE- Hmar genomes to accumulate

deleterious mutations, that resulted in maladaptation, and ultimately limited their beneficial contributions to ticks.

In the context of FLE genome degradation, co-infection with *Midichloria* prevents the breakdown and collapse of nutritional symbiosis in *H. marginatum*; that is, the *Midichloria* genome contains an intact biotin biosynthesis operon, and thus compensates for genome decay of FLE. Inversely, *Midichloria* has only partial pathways for riboflavin and folate and hence cannot meet the nutritional needs of *H. marginatum* for these B vitamins. Consequently, the co-infection of FLE with *Midichloria* forms a cooperative system essential for the nutritional symbiosis in *H. marginatum*. In addition, complementary heme synthesis genes are present in the FLE and *Midichloria* genomes, and these symbionts may be an additional source of heme, perhaps conferring a fitness advantage to *H. marginatum* during periods of starvation. In animals, heme is essential for the function of all aerobic cells, playing an important role in controlling protein synthesis and cell differentiation. Complete biosynthetic heme gene pathways exist in most animals, but not in ticks that need an exogenous source of heme (*Gulia-Nuss, 2016*; *Jia et al., 2020*). Vertebrate blood is usually considered to be their unique source for this essential cofactor (*Perner et al., 2016*), however ticks spend most of their life off host, and thus the presence of heme genes in FLE and *Midichloria* genomes may also indicate symbiont provisioning of heme in *H. marginatum*.

The co-localization of FLE and *Midichloria* in tick ovaries suggests that vertical transmission has led to their stable co-inheritance through tick generations, linking their evolutionary fate, as observed in other symbiotic systems of arthropods (*McCutcheon et al., 2019*; *Vautrin and Vavre, 2009*). This co-transmission fidelity creates the conditions under which cooperative interactions can emerge. The stable coexistence of FLE and *Midichloria* genomes makes various genes redundant, as we observed for B vitamin and heme synthesis pathways. Furthermore, the covariance of FLE and *Midichloria* densities in ticks is positive, suggesting that these two symbionts are not in direct competition but rather cooperative. However, FLE is about 100-times more abundant than *Midichloria*, which can be suggestive of competition. Indeed, negative interactions among these endosymbionts can occur because their tick host provides a limited amount of resources and space. Under this hypothesis, the density of *Midichloria* can be depressed by FLE, which could be more competitive. Regardless of the issue of this interaction, FLE and *Midichloria* depend on each other since their host tick would not survive without their combined production of B vitamins. The asymmetrical abundance of FLE and *Midichloria* may have further implications in nutritional endosymbiosis. First, it suggests that, at lower abundance, *Midichloria* can fully bridge the biotin requirements for *H. marginatum*. Second, it could imply that biotin (produced by *Midichloria*) is required at much lower abundance than riboflavin (FLE) and folate (FLE and *Midichloria*), or that *Midichloria* could over-express its biotin genes, possibly thanks to the streamlined structure of the biotin operon, and finally produce biotin at the same abundance than riboflavin and folate.

While the nutritional association of FLE with *Hyalomma* species has persisted for millions of years (*Azagi et al., 2017*), the nutritional role of *Midichloria* seems more recent in this tick genus. Co-infection with *Midichloria* was reported in at least nine *Hyalomma* species (*Supplementary file 1d*), but in most species, such as *H. excavatum* and *H. impeltatum* (*Azagi et al., 2017*; *Selmi et al., 2019*), *Midichloria* is present at much lower frequencies than would be expected for an obligate nutritional mutualist, suggesting that it is instead a facultative (i.e., not essential) association. Interestingly, the *Hyalomma* species apparently devoid of *Midichloria*, *H. asiaticum*, harbors an FLE with intact biotin, folate, and riboflavin synthesis pathways (*Buysse and Duron, 2021*). In *H. marginatum*, the decay of the FLE genome may have been accelerated by the presence of *Midichloria* and the redundancy of biotin biosynthesis, a process also reported in sap-feeding insects (*Husnik and McCutcheon, 2016*; *Manzano-Marín et al., 2020*). While it is clear that two events have led to the current status (loss of biotin genes and acquisition of *Midichloria*), the order by which these happened is unclear. *Midichloria* could have rescued a decaying symbiosis, or could have led, with its entrance in the system, to relaxation of selective constraints on FLE. Based on the common presence of *Midichloria* at lower frequencies in other *Hyalomma* species, a scenario of gradual parallel occurrence of these two events cannot be ruled out.

Our analyses also indicated that the biotin operon detected in the *Midichloria* of *H. marginatum* has an exogenous origin, being acquired following an ancient lateral gene transfer pre-dating the divergence between the *Midichloria* of *H. marginatum* and the one of *I. ricinus*. Accumulating

genomic sequences indicate that this streamlined biotin operon (denoted in some studies as the Biotin synthesis Operon of Obligate intracellular Microbes—BOOM *Driscoll et al., 2020*) has experienced transfers between distantly related endosymbionts inhabiting arthropod cells including *Wolbachia* (*Driscoll et al., 2020*; *Gerth and Bleidorn, 2016*; *Ju et al., 2020*; *Nikoh et al., 2014*), *Rickettsia* (*Gillespie et al., 2012*), *Cardinium* (*Penz et al., 2012*), and *Legionella* (*Ríhová et al., 2017*). Remarkably, the invasive nature of this streamlined biotin operon has contributed not only to the emergence of novel nutritional symbioses in *H. marginatum* with *Midichloria*, but also in other blood feeders, such as bedbugs with *Wolbachia* (*Nikoh et al., 2014*) and rodent lice with *Legionella* (*Ríhová et al., 2017*), or in sap feeders as white flies with *Cardinium* (*Santos-Garcia et al., 2014*) and planthoppers with *Wolbachia* (*Ju et al., 2020*). Until recently, lateral gene transfer was thought to be rare in intracellular symbionts, as they reside in confined and isolated environments. However, co-infections of unrelated endosymbionts within the same host cell have created freely recombining intracellular bacterial communities, a unique environment previously defined as the 'intracellular arena' (*Kent and Bordenstein, 2010*). The detection of the streamlined biotin operon in *Wolbachia*, *Rickettsia*, and *Cardinium*—three of the most common intracellular symbionts of arthropods (*Duron et al., 2008*)—corroborates the role of such 'intracellular arenas' in the evolutionary dynamics of nutritional symbioses, especially as the biotin operon can be found on plasmids (*Gillespie et al., 2012*).

The macroevolutionary and ecological consequences of acquiring nutritional symbionts are immense, ticks would simply not exist as strict blood feeders without such mutualistic interactions (*Duron and Gottlieb, 2020*). Our investigations corroborate that nutritional symbionts of ticks have all converged towards an analogous process of B vitamin provisioning. While past investigations showed that this evolutionary convergence consists in severe degeneration of symbiont genomes accompanied by preservation of biotin, folate, and riboflavin biosynthesis pathways (*Buysse and Duron, 2021*; *Duron and Gottlieb, 2020*), nutritional endosymbiosis in *H. marginatum* is influenced by a more complex web of interactions resulting in a dynamic mosaic of symbiont combinations and offering novel evolutionary possibilities.

## Materials and methods

### Tick collection, DNA extraction, and molecular screening

Adult females of *H. marginatum* were collected from domestic animals from three geographical regions (Italy, Spain, and Israel) (*Figure 1*). Whole body of one female from Italy was processed for metagenome sequencing, while ovaries from females collected in Spain (n=10) and Israel (n=10) were used for metagenome sequencing (and for qPCR in the case of samples from Spain). For each sample, total DNA was extracted using either the DNeasy Blood and Tissue Kit (QIAGEN) or the NucleoSpin Plant II Kit (Macherey-Nagel).

A molecular screening was performed by qPCR (iQ5 system, Bio-Rad) on Spanish samples to evaluate the bacterial load of FLE and *Midichloria* symbionts in ovaries. For each specimen, three SYBR Green-based qPCR (Bio-Rad) assays were performed: (i) a specific amplification for the *H. marginatum cal* gene, (ii) a specific amplification for the FLE *rpoB* gene, and (iii) a specific amplification for the *Midichloria gyrB* gene. Thermal protocols and primers are listed in *Supplementary file 1e*.

### Genome sequencing, assembly, and annotation

Metagenomes were sequenced on Illumina HiSeq 2500, after a library preparation performed with Nextera XT. For the Italian and the Spanish samples, the raw paired-end reads quality was evaluated with FastQC (*Andrews, 2010*). Paired-end reads were trimmed via Trimmomatic (v0.40) (*Bolger et al., 2014*) and then assembled using SPAdes (v3.10.0) *Bankevich et al., 2012* following the Blobology pipeline (*Kumar et al., 2013*) (thus separating reads according to G+C content, coverage, and taxonomic annotation). Discarding of contaminating sequences and refining of the assemblies were performed manually through the analysis of the assemblies' graphs with Bandage (v0.8.1) (*Wick et al., 2015*). Bacterial reads were separated from the tick reads, allowing the assembling of different bacterial genomes ('filtered' assemblies). For the Israeli sample, reads were trimmed using Cutadapt (v1.13) (*Martin, 2011*) and contaminants' reads were removed through a mapping against Coliphage phi-X174 (GenBank: NC_001422) and *Staphylococcus aureus* (NC_010079) genomes with BWA (v0.7.17-r1188) (option mem) and SAMtools (v1.10) (*Li and Durbin, 2009*). Remaining reads

were further assembled using SPAdes (v3.11.1) (*Bankevich et al., 2012*) with default parameters. These 'raw' assembled scaffolds were compared to different reference genomes separately (FLE of *O. moubata* FLEOm, *M. mitochondrii*, and *R. aeschlimannii*; *Supplementary file 1b*) to discard unspecific scaffolds using RaGOO (v1.02) (*Alonge et al., 2019*), and, for each reference genome separately, the assignation of the remaining scaffolds ('filtered' assembly) was confirmed using the online NCBI BLAST tool (https://blast.ncbi.nlm.nih.gov/Blast.cgi). The quality assessment and the completeness of the 'filtered' assemblies from all *H. marginatum* metagenomes were estimated using QUAST (v4.6.3) (*Gurevich et al., 2013*) and miComplete (v1.1.1) (*Hugoson et al., 2020*) with the --hmms Bact105 setting. The newly obtained genomes were annotated using Prokka (v1.13.1) (*Seemann, 2014*) with default parameters. The new *Midichloria* genomes were annotated using a manually revised annotation of *M. mitochondrii* (*Supplementary file 1b*) as a reference.

## Genomic content, structure, and metabolic pathways analyses

The FLE and *Midichloria* whole-genome alignments were obtained separately using Mauve (*Darling et al., 2004*). Previously, contigs from newly obtained genomes were reordered with Mauve. Graphical representations of newly sequenced genomes were produced using CGView (v1.5) (*Stothard and Wishart, 2005*). Genes involved in biosynthesis pathways (B vitamins and heme), FPI, and mismatch repair systems (*mutSL* and *uvrABCD*) were retrieved using BLASTn, BLASTp, and tBLASTx on both newly obtained and reference FLE and *Midichloria* genomes.

To identify mobile elements, we used ISEScan (*Xie and Tang, 2017*) to predict insertion sequences and PhySpy (*Akhter et al., 2012*) for prophages detection. Pseudogenes prediction was performed using Pseudofinder (version downloaded in date April 1, 2020) (*Syberg-Olsen et al., 2020*) on each genome, including newly obtained and reference genomes of FLE and *Midichloria* (*Supplementary file 1b*). The FLE and *Midichloria* data sets were filtered excluding pseudogenes from the subsequent analyses. Clusters of orthologous groups of proteins (COGs) were predicted using the NCBI pipeline (*Galperin et al., 2015*) and COGs presence was then plotted with an in-house R script. Then, two different data sets were created: (i) a FLE data set including the three newly sequenced FLE-Hmar genomes and other published FLE genomes; and (ii) a *Midichloria* data set including the three newly sequenced Midi-Hmar genomes and *M. mitochondrii*. For each data set, SCOs were identified using OrthoFinder (v2.3.12) (*Emms and Kelly, 2019*), then SCO lists were retrieved on a R environment (v3.6.2) to build Venn diagrams using the 'VennDiagram' R package (*Chen and Boutros, 2011*).

To check if whether B-vitamin biosynthesis genes of other origin than FLE, *Midichloria*, and *Rickettsia*, we used the following procedure: we predicted the genes present in the preliminary assemblies of the three tick metagenomes using prodigal (using the setting '-p meta') and annotated them with GhostKOALA (*Kanehisa et al., 2016*). Indeed, for biotin, we checked for the presence of the *bioA*, *bioD*, *bioC*, *bioH*, *bioF*, and *bioB* genes searching for 'K00833, K01935, K02169, K02170, K00652, and K01012' sequences in the annotations, setting a cutoff to exclude contigs shorter than 500 bp or with a kmer coverage below 4. The hits above cutoff were manually examined with BLAST for functional and taxonomic classification.

## Phylogenomics

The FLE, *Midichloria*, and *Rickettsia* data sets were filtered excluding the pseudogene sequences and were then used for the phylogenetic analyses. For each symbiont data set, SCOs were identified using OrthoFinder (v2.3.12) (*Emms and Kelly, 2019*) and then aligned with MUSCLE (v3.8.31) (*Edgar, 2004*). Poorly aligned positions and divergent regions were removed by Gblocks (v0.91b) (*Castresana, 2000*) prior to concatenation of individual alignments using an in-house script. Finally, the most suitable evolutionary model was determined using modeltest-ng (v0.1.5) (*Darriba et al., 2020*) according to the Akaike information criterion and maximum likelihood (ML) trees were inferred using RAxML (v8.2.9) (*Stamatakis, 2014*) with 1000 bootstrap replicates.

## Biotin operon phylogeny and structure

Nucleic and protein sequences of the six genes of the biotin biosynthesis pathway were retrieved using BLASTn, BLASTp, and tBLASTx on all *Midichloria* genomes. Sequences from described streamlined biotin operons and sequences from biotin genes from other symbionts (retrieved from GenBank) were included in the data set (details in *Supplementary file 1c*). All protein sequences of the six

biotin genes were manually concatenated and aligned with MAFFT (v7.450) (*Katoh et al., 2002*) and ambiguous positions were removed using trimAl (v1.2rev59) (*Capella-Gutiérrez et al., 2009*). The appropriate substitution model, and then ML-tree, were determined as described above. Meanwhile, 16S rRNA sequences from all organisms included in this data set were retrieved from GenBank and aligned with MEGA software. The Gblocks program with default parameters was used to obtain non-ambiguous sequence alignments. Phylogenetic analyses were performed with MEGA software, that is, determination of the appropriate substitution model using Akaike information criterion, ML-tree inference. Clade robustness was assessed by bootstrap analysis using 1000 replicates.

Contigs carrying the streamlined biotin operon were retrieved from newly obtained and reference genomes of different intracellular bacteria. Contigs were annotated using Prokka (v1.13.1) (*Seemann, 2014*) with default parameters. Prokka GenBank outputs were imported in R (v3.6.2) and analyzed using the 'genoplotR' R package (*Guy, 2010*). Sequences of *ribH* and *folD* genes from bacteria carrying the streamlined biotin operon and from other bacteria were secondarily separately analyzed. Phylogenetic analyses on these two genes were performed as described above for biotin genes.

## Bibliographic survey and phylogeny of *Hyalomma*

To access the associations between *Hyalomma* species with FLE and *Midichloria*, we performed an extensive bibliographic survey of published studies. For *Hyalomma* species for which presence/absence of FLE and *Midichloria* is documented, we retrieved cytochrome c oxidase I (*cox1*) gene sequences from GenBank and further performed an ML phylogenetic analysis with MEGA software as described above.

## Statistical analyses

Statistical analyses were carried out using the R statistical package (v3.6.2) and the 'lme4' R package (*Bates et al., 2015*). The FLE load was analyzed using linear mixed-effects models on a data set containing loads measured in ovaries from adult engorged ticks. The normality of the data distribution was tested through a Shapiro-Wilk test with a 95% confidence interval. Outliers were removed from data sets. To build models, *Midichloria* load was fitted as a fixed explanatory variable. The best-fitted model was determined with the following procedure: the maximal model, including all higher-order interactions, was simplified by sequentially eliminating interactions and non-significant terms to establish a minimal model. A likelihood ratio test (using a chi-square distribution and a p-value cutoff of 0.05) was performed to establish the difference between sequential models. The comparison between the densities of FLE and *Midichloria* of *H. marginatum* was tested by a paired t-test.

## Data and code availability

The bacterial genomes obtained in this study were deposited in European Nucleotide Archive (ENA) under the project PRJEB44779 and accession numbers GCA_910592745 (FLE-Hmar-ES), GCA_910592815 (FLE-Hmar-IL), GCA_910592705 (FLE-Hmar-IT), GCA_910592695 (Midi-Hmar-ES), GCA_910592725 (Midi-Hmar-IL), GCA_910592865 (Midi-Hmar-IT), and GCA_910592765 (RAES-Hmar-IT). Metagenome reads were deposited in ENA under accession numbers ERR7255744 (Hmar-ES), ERR7255745 (Hmar-IL), and ERR7255743 (Hmar-IT). Command lines used for the different genomic analyses are available on GitHub (https://github.com/mariebuysse/Hmar-2021; *Buysse et al., 2021*).

## Acknowledgements

The authors are grateful to Tal Azagi for tick collection and extraction and Inbar Plaschkes for initial sequence analysis of the Israeli sample, to Justine Grillet, Valentina Serra, and Catherine Hartley for technical support during molecular analyses. This work benefited from (1) an international 'Joint Research Projects' grant (EVOSYM) co-managed by the Ministry of Science, Technology and Space (Israel) and the Centre National de la Recherche Scientifique (CNRS, France), from (2) The Israel Science Foundation (ISF Grant no. 1074/18), from (3) the Human Frontier Science Programme Grant RGY0075/2017 to DS, and the Italian Ministry of Education, University and Research (MIUR): Dipartimenti di Eccellenza Programme (2018–2022) – Department of Biology and Biotechnology 'L. Spallanzani,' University of Pavia to DS, and from (4) the French Agence Nationale de la Recherche (ANR-21-CE02-0002).

## Additional information

### Funding

| Funder | Grant reference number | Author |
|---|---|---|
| Ministry of Science, Technology and Space | EVOSYM | Yuval Gottlieb |
| Centre National de la Recherche Scientifique | EVOSYM | Olivier Duron |
| Israel Science Foundation | 1074/18 | Yuval Gottlieb |
| Human Frontier Science Program | RGY0075/2017 | Davide Sassera |
| Ministry of Education, University and Research | | Davide Sassera |
| Centre National de la Recherche Scientifique | ANR-21-CE02-0002 | Olivier Duron |

The funders had no role in study design, data collection and interpretation, or the decision to submit the work for publication.

### Author contributions

Marie Buysse, Anna Maria Floriano, Formal analysis, Investigation, Writing – original draft, Writing – review and editing; Yuval Gottlieb, Davide Sassera, Olivier Duron, Conceptualization, Writing – original draft, Writing – review and editing; Tiago Nardi, Formal analysis, Investigation, Writing – review and editing; Francesco Comandatore, Data curation, Formal analysis, Investigation; Emanuela Olivieri, Alessia Giannetto, Ana M Palomar, Benjamin L Makepeace, Chiara Bazzocchi, Alessandra Cafiso, Data curation, Investigation

### Author ORCIDs

Marie Buysse http://orcid.org/0000-0002-8160-2470
Tiago Nardi http://orcid.org/0000-0002-1248-9873
Ana M Palomar http://orcid.org/0000-0002-5461-5874

### Decision letter and Author response

Decision letter https://doi.org/10.7554/eLife.72747.sa1
Author response https://doi.org/10.7554/eLife.72747.sa2

## Additional files

### Supplementary files

• Supplementary file 1. *Supplementary file 1* contains tables with relevant information that was needed for the genome assemblies and their following analyses, including a summary of the assembly information and quality analyses of the newly sequenced genomes. *Supplementary file 1* presents as well the genes and the primers used for qPCR assays conducted for relative quantification of FLE and *Midichloria*.

• Transparent reporting form

### Data availability

The bacterial genomes obtained in this study were deposited in European Nulceotide Archive (ENA) under the project PRJEB44779 and accession numbers GCA_910592745 (FLE-Hmar-ES), GCA_910592815 (FLE-Hmar-IL), GCA_910592705 (FLE-Hmar-IT), GCA_910592695 (Midi-Hmar-ES), GCA_910592725 (Midi-Hmar-IL), GCA_910592865 (Midi-Hmar-IT) and GCA_910592765 (RAES-Hmar-IT). Metagenome reads were deposited in ENA under accession numbers ERR7255744 (Hmar-ES), ERR7255745 (Hmar-IL) and ERR7255743 (Hmar-IT). Command lines used for the different genomic analyses are available on GitHub (https://github.com/mariebuysse/Hmar-2021; copy archived at swh:1:rev:a459aeecb87d3fae17cf6ab8317654feadb62dba).

The following dataset was generated:

| Author(s) | Year | Dataset title | Dataset URL | Database and Identifier |
|---|---|---|---|---|
| Buysse M, Floriano AM, Gottlieb Y, Nardi T, Comandatore F, Olivieri E, Giannetto A, Palomar S, Makepeace B, Bazzocchi C, Cafiso A, Sassera D, Duron O | 2021 | Dual bacterial symbiosis ensures ticks' nutrition through metabolic complementation | http://www.ebi.ac.uk/ena/browser/view/PRJEB44779 | European Nucleotide Archive, PRJEB44779 |

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
