## [Editor Report]

This manuscript analyses endosymbionts harbored by field collected ticks in Spain, Italy and Israel. Using metagenomics, the authors infer the evolutionary history of a newly discovered tripartite interaction between *Hyalomma marginatum* (tick), *Franscisell*a-like endosymbiont (FLE) (ancestral co-obligate endosymbiont) and a *Midichloria* symbiont (more recent co-obligate endosymbiont). This is the first description of such co-obligate symbiotic association in ticks, which usually harbor only one obligate endosymbiont.

---

## [Decision Letter]

**Decision letter after peer review:**

Thank you for submitting your article "A dual endosymbiosis drives nutritional adaptation to hematophagy in the invasive tick *Hyalomma marginatum*" for consideration by *eLife*. Your article has been reviewed by 3 peer reviewers, including Bruno Lemaître as Reviewing Editor and Reviewer #1, and the evaluation has been overseen by Patricia Wittkopp as the Senior Editor. The following individuals involved in review of your submission have agreed to reveal their identity: Sukanya Narasimhan (Reviewer #2); Alejandro Manzano-Marin (Reviewer #3).

This manuscript analyses endosymbionts harbored by field collected ticks in Spain, Italy and Israel. Using metagenomics, the authors infer the evolutionary history of a newly discovered tripartite interaction between Hyalomma marginatum (tick), Franscisella-like endosymbiont (FLE) (ancestral co-obligate endosymbiont) and a Midichloria symbiont (more recent co-obligate endosymbiont). This is the first description of such co-obligate symbiotic association in ticks, which usually harbor only one obligate endosymbiont.

Genome assembly and analysis reveal a pseudogenization of FLE genes of the biotin-synthesis operon while the Midichloria operon is functional. Furthermore, the authors provide an analysis of the heme biosynthetic pathway that also probably requires the participation of both endosymbiont genomes to function properly. Authors interpret and discuss these observations in the light of operon and bacterial phylogeny and propose that Midichloria could have been selected as co-obligate by ticks to provision biotin (and heme), as a complement for a degenerating ancestral obligate symbiont.

The main weakness is that the conclusions are predictions based on the the metagenomic analysis of the endosymbiont genomes. Experimental corroboration must await further technological advancements to manipulate these endosymbionts.

Essential revisions

1) One of my major concerns of these paper is the data behind the conclusion of the co-obligate status of these two symbionts. I will detail them next:

Role of additional symbionts: The authors state they found additional symbionts such as Rickettsia and actually assembled one draft genome for this bacteria. While I do not suspect this is probably the nutritional symbiont (these bacteria generally lack some essential nutrient pathways), the authors do not at least mention whether or not they found the "degraded" vitamin pathways in the Rickettsia genome. It seems the authors decided a priori this was not a nutritional endosymbiont.

Request: Could the authors at least mention whether or not they found the "degraded" vitamin pathways in the Rickettsia genome? It seems from the manuscript no further consideration of this symbiont as a vitamin provider was done.

2) Localisation and distribution of other symbionts: The authors make the point in lines 196-204 that the different starting samples from the Italian (whole female), Spanish (ovaries), and Israeli (ovaries) populations. While I would also expect to find the nutritional symbionts (mostly) present in the ovaries, other organs were not screened for symbionts (e.g. qPCR or FISH microscopy). How are the Rickettsia, Francisella, and Midichloria bacteria distributed? In general terms, one would expect nutritional symbionts to be localised in very specific tissues and not showing a widespread distribution across the host. However, the authors did not do different organ-specific screenings of the symbionts, which hampers the interpretation of the presence/absence of the Francisella/Midichloria symbionts in ovaries.

Request: In order to assess the distribution and possible specific localisation of Midichloria in the tick body,other organs need to be screened for symbionts (e.g. qPCR or FISH microscopy). If finding Midichloria and Francisella in the ovaries is to be used as further evidence for the nature of the Midichloria symbiont, please provide further evidence for the absence or presence of these symbionts in other tick organs.

3) Pseudogenisation in B vitamin genes: While missing genes are not necessarily a good indicator or lack of them in a genome, namely when this genome is fragmented, pseudogenes are a very good indicator of this function not being coded for in the draft genome. In line 218, the authors state that the pseudogenes of bioD, bioC, bioF, and bioB are inactivated by a frameshift. My question is which kind of frameshift? This is because as it has been demonstrated in Tamas et al., 2008 (10.1073/pnas.0806554105), where the authors elegantly showed that infidelity at poly(A) tracks rescues pseudogenes in endosymbionts. Same holds true for the hem genes. Are they also pseudogenised by a frameshift? I cannot further assess this myself due to the unavailability of the genomes. An additional comment comes from the bioA gene missing in Francisella or Riboflavin in Midichloria. While these genes are not present in other tick genomes, the authors made no effort in locating them in H. marginatum (e.g. PCR or searches in the host data generated from sequencing). The possibility of these genes being in the host cannot be discarded, especially in relation to my following point regarding the "bibliographic survey".

Request: Since the authors did not make the genomes available for review, I would kindly ask that they either release them so that the reviewers can assess them, or at the minimum provide links to download them. Given that the authors have metagenomic data of three populations of the tick species, why was no effort put toward identifying these genes in the tick genome (through PCR or by analysing the rest of the scaffolds of the metagenomic assembly)? While other tick species do not have these genes, the particular tick species referred to in this work has not been analysed. In addition, please provide a supplementary figure showing the arrangement of the mentioned genes/pseudogenes in Figure 4. This gives a better impression of any other architectural or other changes in the arrangement or disposition of these genes/pseudogenes.

4. Bibliographic survey: The authors make reference to a bibliographic survey performed to assess the co-existence of Francisella and Midichloria in other studies. Namely, the authors state that "Of 10 Hyalomma species with sufficient published data, all were found infected by FLE, and 9 by Midichloria (Table S4)" and that "Mapping of FLE and Midichloria onto the Hyalomma phylogenetic tree showed that co-infection by FLE and Midichloria is widespread across all Hyalomma lineages". Looking at Table S4, I would argue in fact that in most studies, the authors only report on one of the symbionts, and in most of those where both symbionts where screened for, only one was found (e.g. Azagi et al., 2017, Buysse and Duron 2021, Ravi et al., 2018, Elbir et al., 2019, Alreshidi et al., 2020, Duron et al., 2017). Therefore, I would argue that the bibliographic survey strongly supports the view that only one symbiont tends to be present whenever this species is analysed. Therefore, I find Figure S7 somewhat deceiving when matching it up with Table S4, given that it gives the impression that the bibliographic survey found that these two symbiont have always been found co-existing in H. marginatum, when in fact they have very rarely been found co-existing, but rather they have been found independently on different studies, with some suggesting only one of the two symbionts is housed by H. marginatum.

---

## [Author Response]

Essential revisions1) One of my major concerns of these paper is the data behind the conclusion of the co-obligate status of these two symbionts. I will detail them next:Role of additional symbionts: The authors state they found additional symbionts such as Rickettsia and actually assembled one draft genome for this bacteria. While I do not suspect this is probably the nutritional symbiont (these bacteria generally lack some essential nutrient pathways), the authors do not at least mention whether or not they found the "degraded" vitamin pathways in the Rickettsia genome. It seems the authors decided a priori this was not a nutritional endosymbiont.Request: Could the authors at least mention whether or not they found the "degraded" vitamin pathways in the Rickettsia genome? It seems from the manuscript no further consideration of this symbiont as a vitamin provider was done.

This is an important point and we have now included an analysis of the B vitamin pathways present in the *Rickettsia* genomes (lines 257-263; Figures 4 and Figure 4—figure supplement 4). As expected, we found no evidence that the RAES-Hmar *Rickettsia* are nutritional symbionts in *H. marginatum*: They harbor an incomplete biotin pathway, having only *bioC*, *bioH,* and *bioB*, and an incomplete folate pathway, having *folE*, *folP,* and *folC* (now visible in the new version of Figure 4). For pantothenic acid, only *ilvE* is present. We did not find genes for other B vitamin pathways. Similar is the situation of the heme biosynthesis (genes present: *gltX*, *hemB*, *hemC*, *hemE*, *hemN,* and *hemH*; Genes absent: *hemA*, *hemL*, *hemD*, *hemY*, and *hemG*). As already highlighted in our manuscript, two other pieces of evidence showed that *R. aeschlimannii* is not a nutritional symbiont necessary for tick survival: (1) The bacterium was not detected in the metagenome obtained from ovaries of Spanish specimens of *Hyalomma marginatum* while *Francisella* and *Midichloria* were consistently present (lines 147-151), (2) The bacterium was very similar to the *R. aeschlimannii* strain MC16, previously isolated from *H. marginatum* and identified as a human pathogen (lines 220-224, see also Figure 1—figure supplement 1). Previous studies also showed that *Rickettsia* infection frequencies never exceed ca. 50% in *H. marginatum* field populations (e.g. Azagi et al., Applied and Environmental Microbiology, 2017, DOI: 10.1128/AEM.01302-17) (lines 217-220). Obviously, the RAES-Hmar *Rickettsia* is the pathogen *R. aeschlimannii*.

2) Localisation and distribution of other symbionts: The authors make the point in lines 196-204 that the different starting samples from the Italian (whole female), Spanish (ovaries), and Israeli (ovaries) populations. While I would also expect to find the nutritional symbionts (mostly) present in the ovaries, other organs were not screened for symbionts (e.g. qPCR or FISH microscopy). How are the Rickettsia, Francisella, and Midichloria bacteria distributed? In general terms, one would expect nutritional symbionts to be localised in very specific tissues and not showing a widespread distribution across the host. However, the authors did not do different organ-specific screenings of the symbionts, which hampers the interpretation of the presence/absence of the Francisella/Midichloria symbionts in ovaries.Request: In order to assess the distribution and possible specific localisation of Midichloria in the tick body,other organs need to be screened for symbionts (e.g. qPCR or FISH microscopy). If finding Midichloria and Francisella in the ovaries is to be used as further evidence for the nature of the Midichloria symbiont, please provide further evidence for the absence or presence of these symbionts in other tick organs.

That is another good point. The request is noteworthy, and it will certainly be the subject of a future study. At this time, unfortunately, we cannot perform qPCR or FISH microscopy due to the lack of available adult specimens. There is no laboratory colony for this tick species and, in natura, the life cycle is not favorable since adults will only be available next summer. However, a previous study on *H. marginatum* by Azagi et al., (2017, DOI: 10.1128/AEM.01302-17) showed that high densities of *Francisella* can be observed in Malpighian tubules and ovaries, while infections in salivary glands are scattered. More broadly, the current literature already indicates the high degree of lifestyle specialization toward mutualism of nutritional symbionts in a wide variety of ticks (including members of all major families and genera). Indeed, tick nutritional symbionts have been found primarily in the same two organs, the ovaries and Malpighian tubules (reviewed in Duron and Gottlieb, 2020, DOI: 10.1016/j.pt.2020.07.007; and in Buysse al., 2019, DOI: 10.1016/j.ttbdis.2019.03.014). Ovary infection is consistent with vertical transmission of nutritional symbionts in developing oocytes. A high density of nutritional symbionts in Malpighian tubules supports a nutritional role for these symbionts, as Malpighian tubules are involved in excretion and osmoregulation. A likely hypothesis is that nutritional symbionts use hemolymph compounds to synthesize B vitamins. Other organs, such as the midgut and salivary glands, have also been found sporadically infected, but the significance of this tissue tropism remains poorly understood. We have introduced these key points in the revised version of our manuscript to better explain our reasoning for the presence/absence of *Francisella*/*Midichloria* symbionts in ovaries (lines 76-84, 131-133 and 324-326). We hope that this addition improves the quality of our manuscript on this point.

3) Pseudogenisation in B vitamin genes: While missing genes are not necessarily a good indicator or lack of them in a genome, namely when this genome is fragmented, pseudogenes are a very good indicator of this function not being coded for in the draft genome. In line 218, the authors state that the pseudogenes of bioD, bioC, bioF, and bioB are inactivated by a frameshift. My question is which kind of frameshift? This is because as it has been demonstrated in Tamas et al., 2008 (10.1073/pnas.0806554105), where the authors elegantly showed that infidelity at poly(A) tracks rescues pseudogenes in endosymbionts. Same holds true for the hem genes. Are they also pseudogenised by a frameshift? I cannot further assess this myself due to the unavailability of the genomes. An additional comment comes from the bioA gene missing in Francisella or Riboflavin in Midichloria. While these genes are not present in other tick genomes, the authors made no effort in locating them in H. marginatum (e.g. PCR or searches in the host data generated from sequencing). The possibility of these genes being in the host cannot be discarded, especially in relation to my following point regarding the "bibliographic survey".Request: Since the authors did not make the genomes available for review, I would kindly ask that they either release them so that the reviewers can assess them, or at the minimum provide links to download them. Given that the authors have metagenomic data of three populations of the tick species, why was no effort put toward identifying these genes in the tick genome (through PCR or by analysing the rest of the scaffolds of the metagenomic assembly)? While other tick species do not have these genes, the particular tick species referred to in this work has not been analysed. In addition, please provide a supplementary figure showing the arrangement of the mentioned genes/pseudogenes in FIgure 4. This gives a better impression of any other architectural or other changes in the arrangement or disposition of these genes/pseudogenes.

About data access: We have already deposited our metagenomes on public databases (GenBank) but they are under provisional embargo until the study will be published accession numbers [GCA_910592745 (FLE-Hmar-ES), GCA_910592815 (FLE-Hmar-IL), GCA_910592705 (FLE-Hmar-IT), GCA_910592695 (Midi-Hmar-ES), GCA_910592725 (Midi-Hmar-IL), GCA_910592865 (Midi-Hmar-IT) and GCA_910592765 (RAES-Hmar-IT)]. We follow your request, and share the assemblies with the reviewers through google drive: https://drive.google.com/drive/folders/1X_rd4N_3oQki2DTzsdovrhYdXebvSxPo?usp=sharing

About presence of missing genes in tick genomes: We searched tick metagenomes for other genes involved in B vitamin production. In the two ovarian samples (Hmar-ES and Hmar-IL), we did not find significant hits. In Hmar-IT (a whole tick), we found the genes *bioA*, *bioD*, *bioF*, *bioB* in contigs that we taxonomically attributed to *Staphylococcus*. Previous studies report the presence of *Staphylococcus*, common skin-associated bacteria, on tick exoskeletons midguts, and hypothesize this to be due to contamination, or acquisition during feeding (Greay et al., 2018, DOI10.1186/s13071-017-2550-5). We found no evidence of B vitamin genes inserted in tick genomes. We notified these results in the revised version of manuscript (lines 265-273, including details in the method section at lines 532-540). We also add two further precisions:

1. While the *Midichloria* genomes lack the *folA* gene (encoding for a dihydrofolate reductase), we detected an alternative dihydrofolate reductase gene, *folM* (also known as *ydgB*), that was shown in other bacteria to provide the same metabolic function than *folA* (Giladi et al., 2003, DOI:10.1128/JB.185.23.7015-7018.2003). The presence of *folM*, instead of *folA*, indicates that the folate pathway is complete in *Midichloria* genomes. We notified this in the revised version of our manuscript (lines 249-253 and Figure 4).

2. In *Francisella* genomes of *H. marginatum*, while we previously found no evidence of *bioA* sequence traces, our reexamination of the dataset reveals the presence of a reduced pseudogenized *bioA* fragment in all Hmar metagenomes (now shown in Figure 4 and indicated at line 232-234).

About the nature of frameshift: This is a very interesting point and we carefully check for this. Pseudogenization are due to indels in almost all cases. We included in the revised version of manuscript a novel figure showing the nature of pseudogenization in the biotin synthesis genes, which are indels in most cases (Figure 4—figure supplement 3). This only exception is *bioB*: A stop codon comes from a reading frame shift due to the insertion of one adenine base on a poly(A) zone. We introduce this point in the revised version of the manuscript (lines 235-241 and Figure S4) and indicate that infidelity at poly(A) tracks could rescue *bioB*. However, it does not change our interpretation (FLE cannot produce biotin) since observations of four other biotin synthesis genes confirm that they are true pseudogenes.

4. Bibliographic survey: The authors make reference to a bibliographic survey performed to assess the co-existence of Francisella and Midichloria in other studies. Namely, the authors state that “Of 10 Hyalomma species with sufficient published data, all were found infected by FLE, and 9 by Midichloria (Table S4)” and that “Mapping of FLE and Midichloria onto the Hyalomma phylogenetic tree showed that co-infection by FLE and Midichloria is widespread across all Hyalomma lineages”. Looking at Table S4, I would argue in fact that in most studies, the authors only report on one of the symbionts, and in most of those where both symbionts where screened for, only one was found (e.g. Azagi et al., 2017, Buysse and Duron 2021, Ravi et al., 2018, Elbir et al., 2019, Alreshidi et al., 2020, Duron et al., 2017). Therefore, I would argue that the bibliographic survey strongly supports the view that only one symbiont tends to be present whenever this species is analysed. Therefore, I find Figure S7 somewhat deceiving when matching it up with Table S4, given that it gives the impression that the bibliographic survey found that these two symbiont have always been found co-existing in H. marginatum, when in fact they have very rarely been found co-existing, but rather they have been found independently on different studies, with some suggesting only one of the two symbionts is housed by H. marginatum.

The main issue here is that the studies listed in Supplementary File 1d do not used the same standardized methods, and rarely screened for *Francisella* and *Midichloria*. So, it is difficult to produce a coherent global picture, and we try to be parsimonious: This is why Figure 1—figure supplement 2 could appear somewhat deceiving. However, as we discussed in the manuscript, some obvious evidences exist (lines 414-430). For studies looking for *Francisella*, most found that infection is close to fixation in Hyalomma spp. (eg Azagi et al., 2017, DOI: 10.1128/AEM.01302-17), supporting the hypothesis of an ancestral nutritional symbiont. For studies looking for Midichloria, most found that a medium prevalence, supporting the hypothesis of a facultative symbiont in some species (eg, H. excavatum and H. impeltatum; Azagi et al., 2017, DOI: 10.1128/AEM.01302-17; Selmi et al., 2019, DOI:10.1016/j.micpath.2019.103655). This explains why most of the Hyalomma species are identified as co-infected in Figure 1—figure supplement 3 – and this is true, there are – but for some species (H. excavatum and H. impeltatum) Midichloria is a facultative symbiont while it is co-obligatory symbiont at least in H. marginatum.

About previous studies, you state that ‘most of those where both symbionts where screened for, only one was found’, but this interpretation is not entirely exact and should treated with caution. For instance, in the study by Azagi et al., (2017, DOI: 10.1128/AEM.01302-17), the authors used Francisella specific primers (and primers targeting other microbes) but none targeting Midichloria: So, it is normal that they report only the presence of Francisella while Midichloria was also probably present. In the study by Duron et al., 2017 (DOI: 10.1111/mec.14094), the authors already detected Francisella and Midichloria in H. marginatum and in H. excavatum – and they also detected Francisella in most of the other Hyalomma species. We agree that there is one exception: The study by Buysse and Duron, 2021 in Cell (DOI: 10.1016/j.cell.2021.03.053) showed that only Francisella was present in H. asiaticum metagenomes but it makes sense since the authors also evidenced that the biotin, folate, and riblofavin pathways are complete in this Francisella genome (Midichloria is thus not needed for complementation in this tick species).